

# Virtual Cockpit
## System Zdalnego Sterowania Pojazdami Jeżdżącymi



**Autorzy**: Kacper Gaudyn ⬤ · Michał Pesta ⬤ · Weronika Łoś ⬤ · Daniel Górka ⬤ · Marcin Fedorowicz ⬤

**Opiekun:** Arkadiusz Warzyński

**Streszczenie**

Artykuł przedstawia opracowanie systemu zdalnego sterowania wspierającego autonomiczne pojazdy w sytuacjach krytycznych, gdzie algorytmy AI mogą okazać się niewystarczające. System pozwala operatorowi na efektywne podejmowanie decyzji dzięki niskim opóźnieniom oraz intuicyjnym interfejsom dostępnym w aplikacjach webowej i mobilnej. Opracowane rozwiązanie integruje różnorodne metody sterowania, dostosowane do preferencji użytkownika, oraz wykorzystuje zaawansowane technologie, takie jak Angular i Flutter. System znajduje zastosowanie w zarządzaniu flotą pojazdów autonomicznych, obsłudze pojazdów w zajezdniach oraz w wymagających środowiskach przemysłowych. Dodatkowo, integracja funkcji wizyjnych oraz czujników odległości wspiera operatora w zachowaniu precyzji i bezpieczeństwa. Wyniki testów laboratoryjnych potwierdzają, że system spełnia założenia projektowe, co czyni go elastycznym rozwiązaniem możliwym do wdrożeń w różnych branżach.

# 1 ROZWÓJ OPROGRAMOWANIA

## 1.1 Wstęp

### 1.1.1 Charakterystyka problemu

Rozwój pojazdów autonomicznych to jedno z kluczowych wyzwań współczesnej technologii. Algorytmy sztucznej inteligencji (AI), choć coraz bardziej zaawansowane, wciąż napotykają trudności w radzeniu sobie z nieprzewidywalnymi sytuacjami na drodze. [2] Zdarzenia takie jak nagłe zmiany warunków atmosferycznych, nietypowe zachowania innych uczestników ruchu czy awarie infrastruktury mogą prowadzić do sytuacji, w których algorytmy działają mniej efektywnie lub nie są w stanie podjąć właściwych decyzji. Aby zapewnić bezpieczeństwo i płynność operacyjną pojazdów autonomicznych, konieczne jest opracowanie systemu umożliwiającego człowiekowi zdalne przejęcie kontroli w krytycznych momentach.

Jednym z głównych wyzwań jest zapewnienie niskiego opóźnienia w komunikacji między pojazdem a operatorem, co jest kluczowe w sytuacjach kryzysowych. Problemem jest także skuteczna integracja strumienia wideo o wysokiej jakości z systemem sterowania w czasie rzeczywistym, aby umożliwić precyzyjne manewrowanie w wymagających środowiskach.

### 1.1.2 Cele projektu

Głównym celem projektu jest opracowanie systemu zdalnego sterowania wspierającego pojazdy autonomiczne, który umożliwi operatorowi natychmiastową interwencję w sytuacjach, w których algorytmy AI napotkają trudności.

Cele szczegółowe obejmują:

1. Zapewnienie opóźnienia sterowania mniejszego niż 0,5 sekundy, co pozwala na natychmiastową reakcję operatora i sprawne przekazanie poleceń pojazdowi w sytuacjach krytycznych,

2. Udostępnienie strumienia wideo o opóźnieniu mniejszym niż 1 sekunda, co umożliwia operatorowi skuteczne monitorowanie otoczenia pojazdu i podejmowanie trafnych decyzji,

3. Udostępnienie funkcji szybkiego połączenia z pojazdem autonomicznym będącym w ruchu, co zapewnia możliwość przejęcia kontroli w czasie rzeczywistym.

Różnice w założeniach dotyczących opóźnień sterowania i transmisji obrazu wynika z różnic w wymaganiach dotyczących przetwarzania i przesyłania danych. Komendy sterowania mają niewielki rozmiar i wymagają jedynie minimalnego czasu na ich przesłanie, co pozwala na niemal natychmiastowe reakcje systemu. Z kolei przesyłanie obrazu wideo jest procesem bardziej złożonym, gdyż obejmuje kilka etapów przetwarzania. Po stronie nadawczej obraz jest najpierw enkodowany, tj. kompresowany do odpowiedniego formatu w celu zmniejszenia jego rozmiaru. Następnie dane te są przesyłane przez sieć, a po stronie odbiorczej odkodowywane, konwertowane do postaci bitmapy i wyświetlane na ekranie operatora. Każdy z tych etapów wprowadza niewielkie opóźnienie, co może zwiększyć całkowity czas transmisji.

Ze względu na większe wymagania przepustowości i przetwarzania obrazu początkowo ustalono dla transmisji wideo tolerancję opóźnienia wynoszącą do 1 sekundy, co pozwalało zapewnić płynność i niezawodność przesyłu. Dzięki optymalizacji systemu i zastosowaniu zaawansowanych algorytmów kompresji oraz dekompresji udało się jednak osiągnąć porównywalnie niskie opóźnienia zarówno w przesyłaniu obrazu, jak i w sterowaniu, co zwiększyło spójność i skuteczność działania całego systemu.

### 1.1.3 Znaczenie projektu w praktyce

Projekt odgrywa kluczową rolę zarówno w kontekście biznesowym, jak i technologicznym, odpowiadając na aktualne wyzwania związane z funkcjonowaniem pojazdów autonomicznych. Jednym z najistotniejszych problemów jest brak elastyczności systemów autonomicznych w obliczu nietypowych sytuacji, takich jak nagłe awarie infrastruktury, nieprzewidywalne zachowania innych uczestników ruchu czy trudne warunki atmosferyczne. Tego rodzaju sytuacje wymagają szybkiej interwencji, której algorytmy autonomiczne mogą nie być w stanie zrealizować w sposób skuteczny.

Nasze rozwiązanie umożliwia zdalną interwencję operatora w krytycznych momentach, co zwiększa bezpieczeństwo i minimalizuje ryzyko przestojów operacyjnych. Możliwość jednoczesnego zarządzania dużą liczbą pojazdów przez niewielką liczbę operatorów pozwala na optymalizację kosztów, zwiększając efektywność zarządzania flotą. Ponadto zastosowanie zaawansowanych technologii, takich jak przetwarzanie danych w czasie rzeczywistym oraz intuicyjne interfejsy użytkownika, usprawnia integrację rozwiązania z istniejącą infrastrukturą.

Projekt cechuje się również dużą uniwersalnością, co pozwala na jego zastosowanie w różnych środowiskach operacyjnych. Oprócz autonomicznego ruchu drogowego może być wykorzystywany w zarządzaniu zajezdniami autobusowymi, gdzie niezbędne jest precyzyjne sterowanie pojazdami w ograniczonej przestrzeni. Technologię można także łatwo dostosować do innych typów pojazdów, takich jak ciężarówki, maszyny budowlane czy pojazdy używane w transporcie publicznym, co otwiera szerokie możliwości rozwoju i wdrożeń w różnorodnych sektorach.

### 1.1.4 Przykłady potencjalnych zastosowań systemu

1. **Zarządzanie flotą pojazdów autonomicznych w przedsiębiorstwach**
   System może wspierać centralne zarządzanie flotą, umożliwiając szybką reakcję na sytuacje kryzysowe lub awarie pojazdów. Zdalna kontrola pozwala operatorom podejmować odpowiednie działania bez konieczności fizycznej obecności w miejscu zdarzenia, co znacząco zwiększa efektywność operacyjną.

2. **Obsługa autobusów na terenie zajezdni**
   Potencjalne zastosowanie systemu obejmuje sterowanie autobusami w zajezdniach, co umożliwia ich precyzyjne przemieszczanie, parkowanie oraz przekierowywanie do myjni czy serwisu. Automatyzacja tych procesów prowadzi do zmniejszenia zapotrzebowania na personel i obniżenia kosztów operacyjnych.

3. **Obsługa pojazdów na terenie lotnisk**
   Na lotniskach system może być wykorzystywany do obsługi pojazdów odpowiedzialnych za transport bagaży, pasażerów czy towarów. Zdalne sterowanie zwiększa efektywność operacyjną floty oraz pozwala na ograniczenie liczby osób potrzebnych do obsługi w terenie.

4. **Sterowanie pojazdami w wymagających środowiskach przemysłowych**
   W środowiskach przemysłowych, takich jak fabryki czy kopalnie, system może znaleźć zastosowanie do zdalnego sterowania maszynami autonomicznymi. Takie rozwiązanie pozwala na bezpieczne i wydajne działanie w ekstremalnych warunkach, które mogą stanowić wyzwanie dla standardowych algorytmów.

Podsumowując, wdrożenie systemu zdalnego sterowania dla pojazdów autonomicznych otwiera szerokie możliwości zastosowań w różnych branżach, zwiększając efektywność operacyjną, obniżając koszty i zapewniając większe bezpieczeństwo użytkowania.

## 1.2   Prace powiązane

Sterowanie pojazdami autonomicznymi stanowi aktualny obszar badań i rozwoju, w związku z czym nieustannie powstają nowe, zaawansowane rozwiązania informatyczne wspierające tę dziedzinę. Jednym z takich projektów jest "System bezprzewodowego zdalnego sterowania dla pojazdu autonomicznego", opracowany przez zespół Politechniki Poznańskiej (Piotr Góral, Paweł Pawłowski, Adam Dąbrowski) [5]. Celem tego projektu było stworzenie systemu zdalnego sterowania pojazdem przeznaczonym do prac sadowniczych.

W konkurencyjnym rozwiązaniu sterowanie lokalne realizowane jest przy użyciu przemysłowego sterownika PLC (Siemens S7-1200), podczas gdy zdalna komunikacja odbywa się poprzez aplikację komputerową, napisaną w języku C# w środowisku Visual Studio. Dane między operatorem a pojazdem przesyłane są za pośrednictwem sieci WiFi w standardzie IEEE 802.11n, co umożliwia dwukierunkową wymianę informacji. System ten został zaprojektowany z myślą o stabilności i niezawodności działania w ograniczonym środowisku pracy, co czyni go odpowiednim dla określonych zastosowań, takich jak prace w sadach.

Głównym ograniczeniem tego projektu jest jednak jego wąski zakres funkcjonalności. Aplikacja sterująca dostępna jest wyłącznie na komputery stacjonarne, co znacznie ogranicza elastyczność użytkowania. Brak wsparcia dla urządzeń mobilnych oraz ograniczenie możliwości systemu wizyjnego do pojedynczej kamery utrudniają efektywne reagowanie na dynamiczne zmiany w środowisku.

W porównaniu z tym rozwiązaniem nasz projekt wyróżnia się nowoczesnym i uniwersalnym podejściem, które otwiera znacznie szersze możliwości zastosowania. Dzięki wykorzystaniu technologii Angular i Flutter nasz system obsługuje zarówno platformy webowe, jak i mobilne, co daje operatorom większą elastyczność w dostępie do funkcji sterowania. Operator ma również możliwość wyboru spośród różnych metod sterowania, co pozwala na dostosowanie interfejsu i sposobu kontroli do indywidualnych preferencji, posiadanego sprzętu oraz specyfiki wykonywanych zadań. Dodatkowo zastosowanie Redis i Firebase pozwala na łatwą skalowalność systemu, umożliwiając zarządzanie większą liczbą pojazdów w ramach jednej infrastruktury, co czyni nasze rozwiązanie lepiej dostosowanym do potrzeb kompleksowych systemów flotowych. Co więcej, nasz projekt korzysta z dwóch kamer, co zapewnia szersze pole widzenia i dokładniejszy podgląd otoczenia, a także został wyposażony w czujniki odległości, które wspierają operatora w unikaniu kolizji. Czujniki te nie tylko rejestrują obecność przeszkód, ale również ostrzegają operatora, gdy obiekt znajduje się w niebezpiecznej odległości, co znacząco zwiększa poziom bezpieczeństwa. Takie podejście pozwala na bardziej efektywne manewrowanie pojazdem w trudnych i ograniczonych przestrzeniach, co stanowi istotną przewagę nad systemem konkurencyjnym.

Podsumowując, nasze rozwiązanie oferuje znacznie większą elastyczność, bezpieczeństwo i funkcjonalność, co czyni je bardziej uniwersalnym i przyszłościowym w porównaniu do rozwiązania konkurencyjnego. Dzięki nowoczesnym technologiom oraz integracji zaawansowanych funkcji nasz system lepiej odpowiada na potrzeby dynamicznie rozwijającego się rynku autonomicznych pojazdów.

## 1.3   Rezultaty

System zdalnego sterowania pojazdami jeżdżącymi został pomyślnie zrealizowany, osiągając znaczące rezultaty w zakresie funkcjonalności, wydajności oraz potencjału wdrożeniowego. Wyniki zostały poparte testami w warunkach laboratoryjnych. Dla przejazdu testowego, osiągnięto średnią wartość opóźnienia transmisji wideo na poziomie 0,04 s, natomiast średnia wartość opóźnienia dla przesyłanych instrukcji ruchu wyniosła 0,004 s. Testy zostały przeprowadzone bez rozproszenia, pomijając w ten sposób opóźnienia sieciowe, które są czynnikiem niezależnym od implementacji systemu. Ponadto oprogramowanie zostało objęte testami jednostkowymi, osiągając poziom pokrycia na poziomie ponad 80%. Wyniki przeprowadzonych testów dotyczących opóźnień zostały przedstawione na poniższych wykresach.

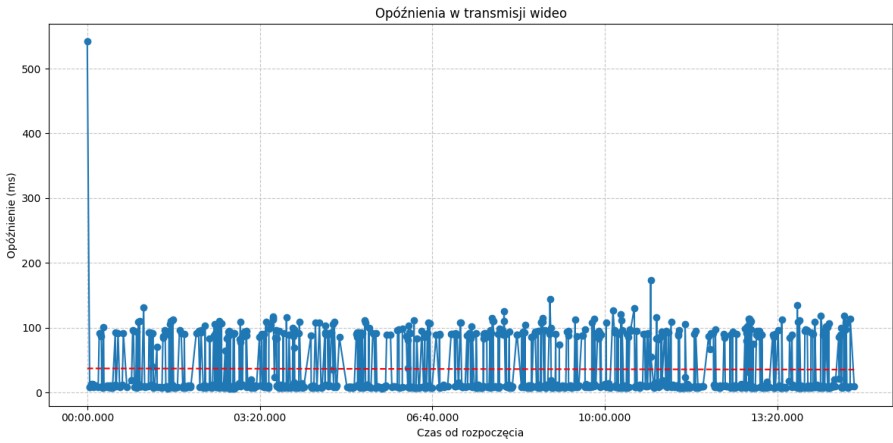

Rysunek 1: Opóźnienia transmisji wideo

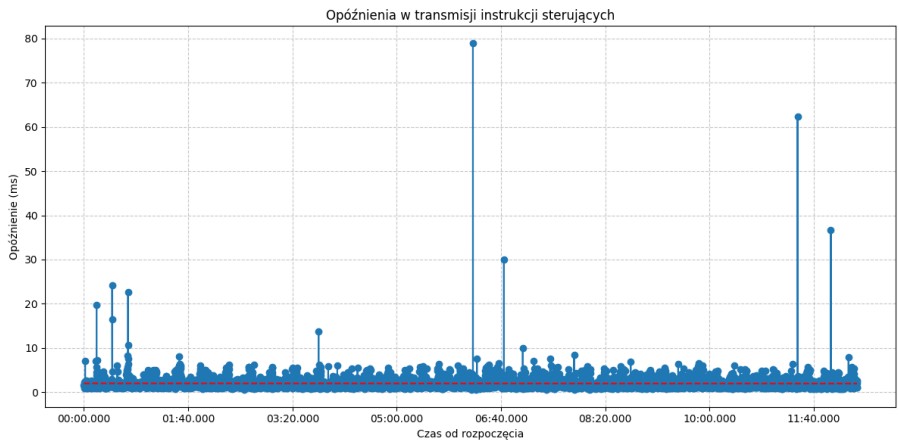

Rysunek 2: Opóźnienia transimsji instrukcji sterujących

### 1.3.1 Zrealizowane funkcjonalności

System został zaimplementowany w postaci zintegrowanego rozwiązania obejmującego aplikację webową oraz mobilną. Główne funkcjonalności obejmują:

1. **Strumieniowanie wideo:**

   · Opracowano kanał komunikacyjny do przesyłania obrazu w czasie zbliżonym do rzeczywistego z kamer pojazdu z częstotliwością 30 klatek na sekundę,

   · Wdrożono wielokanałowy podgląd z dwóch kamer pojazdu jednocześnie,

   · Wydajność systemu została przetestowana, a otrzymane wyniki wykazały średnie opóźnienie transmisji obrazu na poziomie 0,04 sekundy,

   · Zaimplementowano możliwość automatycznego lub ręcznego przełączania podglądu pomiędzy kamerą przednią a tylną, w zależności od potrzeb użytkownika.

2. **Sterowanie pojazdem:**

   · Wdrożono wsparcie dla trzech różnych metod sterowania z poziomu aplikacji mobilnej dostosowanych do preferencji użytkownika:

     − Sterowanie z wykorzystaniem akcelerometru – wykorzystuje akcelerometr wbudowany w urządzenie mobilne. Sensor ten mierzy przyspieszenia o charakterze stałym (grawitacja), zmiennym w czasie (wibracje) oraz quasi-statycznym (pochylenie) na trzech osiach (x, y, z), wyrażanych w metrach na sekundę kwadrat (m/s²) [4]. Dzięki temu poprzez przechylanie urządzenia na boki użytkownik zmienia promień skrętu pojazdu, co umożliwia intuicyjną kontrolę kierunku jazdy. Dodatkowo, na ekranie znajduje się suwak pozwalający na regulację prędkości, w tym jazdę wstecz,

- – Sterowanie za pomocą dwóch suwaków – dedykowane pojazdom wyposażonym w dwie niezależne strony sterowania. Metoda ta umożliwia precyzyjną i niezależną kontrolę każdej ze stron pojazdu,
- – Sterowanie joystickiem wirtualnym – umożliwia użytkownikowi sterowanie pojazdem jedną ręką. Poprzez przesuwanie wirtualnego joysticka na ekranie można kontrolować ruch pojazdu we wszystkich kierunkach, co zwiększa wygodę obsługi.

- · Zaimplementowano wsparcie dla dwóch różnych metod sterowania z poziomu aplikacji webowej dostosowanych do preferencji użytkownika:
  - – Sterowanie za pomocą klawiatury – wykorzystuje standardowe klawisze kierunkowe do kontroli pojazdu. Rozwiązanie to zapewnia szybki i precyzyjny dostęp do sterowania, szczególnie w przypadku urządzeń bez dodatkowych akcesoriów,
  - – Sterowanie za pomocą kierownicy Thrustmaster T-GT II – Kierownica umożliwia precyzyjną kontrolę kierunku, a wbudowane pedały pozwalają na intuicyjną regulację prędkości, hamowania oraz jazdę wstecz. Rozwiązanie to idealnie sprawdza się w przypadku symulacji i bardziej zaawansowanych zastosowań.
- · Wydajność systemu została przetestowana, a otrzymane wyniki wykazały średnie opóźnienie transmisji instrukcji sterowania na poziomie 0,004 sekundy,
- · Zastosowano zabezpieczenie systemu tak aby nastąpiło awaryjne zatrzymanie pojazdu w przypadku problemów sieciowych.

3. **Interfejs operatora:**

- · Wykorzystując framework Angular, zaimplementowano intuicyjny panel sterowania dostępny przez przeglądarkę internetową,
- · Stworzona we framework'u Flutter aplikacja mobilna, umożliwia sterowanie w terenie,
- · Dla obu aplikacji wprowadzono system ostrzegania użytkownika o sytuacjach niebezpiecznych,
- · Zaimplementowano podgląd aktualnej odległości z przedniego czujnika, wzbogacony o wizualne ostrzeżenia informujące o zmniejszającej się odległości w celu zapewnienia bezpieczeństwa użytkownika.

### 1.3.2 Spełnienie celów projektu

Realizacja założeń projektowych zakończyła się pełnym sukcesem, osiągając zarówno cele biznesowe, jak i techniczne. System wykazuje wyjątkową wydajność w kluczowych parametrach operacyjnych. Przeprowadzone testy w warunkach laboratoryjnych, z wykorzystaniem infrastruktury sieci 5G, potwierdziły, że opóźnienia w sterowaniu pojazdem utrzymują się stabilnie poniżej zakładanego progu 0,5 sekundy. Równie imponujące wyniki osiągnięto w przypadku transmisji obrazu - opóźnienia strumieniowania wideo nie przekraczają jednej sekundy, co ma fundamentalne znaczenie dla bezpieczeństwa operacji i skutecznego reagowania w sytuacjach awaryjnych.

Stworzone interfejsy użytkownika, zarówno w formie aplikacji mobilnej, jak i webowej, charakteryzują się wysoką funkcjonalnością i ergonomią. Zapewniają one błyskawiczne nawiązywanie połączenia z pojazdami autonomicznymi, co jest kluczowe w sytuacjach wymagających natychmiastowej interwencji operatora. Natomiast intuicyjny system przełączania między pojazdami znacząco usprawnia pracę operatorów nadzorujących jednocześnie wiele jednostek.

Opracowane rozwiązanie osiągnęło dojrzałość technologiczną umożliwiającą jego implementację w środowiskach testowych. System jest również w pełni przygotowany do dalszej rozbudowy i dostosowywania do specyficznych wymagań klientów. Szeroki wachlarz potencjalnych zastosowań, szczegółowo opisanych w punkcie 2.1.4 dokumentacji, czyni system wszechstronnym narzędziem do zarządzania flotą pojazdów autonomicznych. Jego wdrożenie przekłada się na wymierne korzyści operacyjne - minimalizację przestojów w pracy floty oraz znaczące podniesienie poziomu bezpieczeństwa realizowanych operacji.

# 2 WNIOSKI

## 2.1 Wnioski

Przeprowadzone prace nad systemem zdalnego sterowania dla pojazdów autonomicznych wykazały istotne możliwości rozwoju w dziedzinie wspierania systemów sztucznej inteligencji w krytycznych sytuacjach. Stworzony system spełnia wszystkie założone cele projektowe, oferując niskie opóźnienia w komunikacji, intuicyjne interfejsy użytkownika oraz zaawansowane funkcje zwiększające bezpieczeństwo operacyjne. Kluczowe rezultaty obejmują:

- **Wysoka wydajność techniczna**:
  - Opóźnienia w sterowaniu i transmisji wideo poniżej zakładanych progów umożliwiają skuteczną interwencję w czasie rzeczywistym.
  - Integracja dwóch kamer oraz czujników odległości zapewnia operatorowi pełniejszy wgląd w sytuację na drodze, zwiększając precyzję i bezpieczeństwo działań.

- **Uniwersalność i elastyczność**:
  - Obsługa różnych platform (web i mobile) oraz metod sterowania dostosowanych do potrzeb operatorów czyni system wszechstronnym i łatwo adaptowalnym.
  - Możliwość skalowania systemu pozwala na zarządzanie flotami pojazdów w różnych branżach, takich jak transport publiczny, przemysł czy logistyka.

- **Przewaga technologiczna**:
  - W porównaniu z istniejącymi rozwiązaniami konkurencyjnymi system wyróżnia się większą funkcjonalnością, lepszym dostosowaniem do dynamicznych zmian środowiska oraz bardziej ergonomicznym podejściem do interfejsów użytkownika.

## 2.2 Kierunki rozwoju

Nasze oprogramowanie daje operatorowi możliwość sterowania pojazdem wyposażonym w interfejs pozwalający na obsługę instrukcji sterujących, otrzymywanie strumieni wideo z kamer oraz danych z czujników. Większość pojazdów nie posiada takiego interfejsu w postaci ogólnodostępnej.

W pojazdach nieautonomicznych nie ma interfejsu, który pozwala na przejęcie sterowania przez program - jedynym elementem, który może sterować pojazdem jest kierowca, który wykorzystuje do tego kierownicę, pedały itp. W pojeździe autonomicznym mamy do czynienia z rozszerzonym interfejsem sterowania, z którego - oprócz kierowcy - może korzystać program sterujący.

Jeżeli taki pojazd pozwala na wykorzystanie tego interfejsu w inny sposób, jest wówczas możliwość dodania kolejnego elementu sterującego - programu, który pozwala osobie fizycznej na sterowania samochodu zdalnie. Jeżeli sterowanie zdalne okaże się skuteczne, producenci samochodów autonomicznych mogą przekonać się do wyposażania swoich samochodów w tę możliwość.

### 2.2.1 Kontrolowanie autonomicznych pojazdów w sytuacjach krytycznych

Jesteśmy świadkami rewolucji technologicznej, której głównym podmiotem jest sztuczna inteligencja. Algorytmy SI wykorzystywane są w niemalże każdej dziedzinie, również w codziennym użytku pojazdów. Na ulicach możemy spotkać wiele pojazdów pozwalających kierującemu na oderwanie rąk od kierownicy, co więcej firmy transportowe korzystają z pełnie autonomicznych pojazdów do przewozu osób (np. Waymo - firma transportowa stworzona jako projekt Google. Korzystają oni z samochodów autonomicznych wyposażonych w kamery oraz czujniki. Korzystać z usługi można w Stanach Zjednoczonych - w Los Angeles, San Francisco oraz Phoenix [3]). Kwestią czasu jest wprowadzenie autonomicznych pojazdów (autobusów, tramwajów) w sektorze komunikacji miejskiej.

Rozważmy sytuację, w której algorytm sterujący zawodzi, bądź okazuje się niewystarczająco precyzyjny. W takim przypadku pojazd powinien umożliwić sterowanie osobie fizycznej, niemniej umieszczanie nadzorcy/kierowcy w każdym takim pojeździe nie wydaje się być dobrym rozwiązaniem - w kwestii zarządzania zasobami, wymagany jest wówczas pracownik, którym mógłby być kierowcą w pojeździe nieautonomicznym, a sama idea sterowania przy pomocy algorytmu okazuje się nieopłacalna. Lepszym rozwiązaniem jest wyznaczenie operatora, który ma możliwość zdalnego sterowania pojazdem, wykorzystując dwukierunkowe strumieniowanie danych między pojazdem, a osobą sterującą.

Nasz system umożliwia operatorowi wybór pojazdu oraz rozpoczęcie jego sterowanie, zapewnia widok z kamer, oraz przesyła informację z czujników zamontowanych w pojeździe. Dzięki powyższym możliwościom, osoba fizyczna jest w stanie sprawnie poradzić sobie z sytuacją krytyczną, a pojazd - po opanowaniu sytuacji - może powrócić do trybu autonomicznego.

### 2.2.2 Kontrolowanie pojazdów komunikacji miejskiej w zajezdniach

Po zakończonej trasie, pojazdy komunikacji miejskiej trafiają do zajezdni, w której poddawane są np. ładowaniu/tankowaniu, czy parkowaniu. Przy niektórych aktywnościach, kierowca takiego pojazdu jest odpowiedzialny wyłącznie za przejazd pojazdu z jednego punktu do drugiego.

W takich sytuacjach pojazdami mógłby kierować operator - kierowca może w tym czasie wykorzystać przerwę w prowadzeniu pojazdu. Ograniczy to zmęczenie kierowcy, a co za tym idzie - przyczyni się do zwiększenia bezpieczeństwa w trakcie przejazdu. Według polskiego prawa każdy kierujący powyżej 4,5 godziny ma obowiązek minimum 45-minutowej przerwy w prowadzeniu pojazdu [1]. Operator mógłby przełączać się między pojazdami w zajezdni i zmieniać ich położenie.

Operator mógłby być odpowiedzialny również za kierowanie pojazdami autonomicznymi w zajezdni, wówczas implementacja tras takiego pojazdu nie wymagałaby dodatkowej implementacji poruszania się po zajezdni, co może się przyczynić również do usprawnienia ruchu w zajezdniach z pojazdami autonomicznymi.

## 2.3 Podziękowania

Dziękujemy opiekunowi projektu - dr. inż. Arkadiuszowi Warzyńskiemu za nadzorowanie rozwoju oraz poświęcenie czasu na rzecz pomocy przy problemach napotkanych w trakcie pracy. Składamy również podziękowania dla pracowników Laboratorium technologii usługowych i sieciowych za udostępnienie sprzętu.

## LITERATURA

[1] Kadry Infor. Plan czasu pracy kierowcy autobusu – obowiązek pracodawcy. `https://kadry.infor.pl/kadry/indywidualne_prawo_pracy/czas_pracy/5401020,` `Plan-czasu-pracy-kierowcy-autobusu-obowiazek-pracodawcy.html`, 2022.

[2] Yanjun Huang Jian Zhao Jun Wang, Li Zhang. Safety of autonomous vehicles. `https://` `onlinelibrary.wiley.com/doi/full/10.1155/2020/8867757`, 2020.

[3] © 2019-2024 Waymo LLC. Waymo - autonomous driving technology. `https://waymo.com`.

[4] Z. Mohammed, I. Elfadel, and M. Rasras. Monolithic multi degree of freedom (mdof) capacitive mems accelerometers. *Micromachines*, 9:602, 2018.

[5] Adam DĄBROWSKI Piotr GÓRAL, Paweł PAWŁOWSKI. System bezprzewodowego zdalnego sterowania dla pojazdu autonomicznego, 2019.
