# OpenReview forum: "VirtualCockpit"
_pwr.edu.pl/Wrocław_University_of_Science_and_Technology/2024/ZPI_Day — Wrocław University of Science and Technology 2024 ZPI Day Submission_

### Official Review · Reviewer_rVUm · 2024-12-03
**Virtual Cockpit**

**Confidence:** 2
**Significance Of Results:** 5
**Overall Quality:** 5

**Compliance With Template:**

5: Very High Quality – The article contains all the required sections, which are written in a very detailed, clear, and error-free manner. The structure is professional and meets expectations, and the content adheres to the highest substantive and formal standards.

**Description Of Results:**

5: Very High Quality – The results are described in detail, clearly and comprehensively, supported by thorough evaluation, analysis, and convincing usage examples. The description meets the highest substantive standards.

**Feedback On Consistency:**

Consistency of the project description is very high quality. The problem analysis, presentation of results, and conclusions are consistent and logical.

**Potential For Development:**

Project has high potential for development and has possibilities for further work and practical applications of its results in real world.

**Project Nature Evaluation:**

Virtual Cockpit project exhibit all characteristics of high value engineering work, such as the level of utility, application of technical methods, and technological solutions.

**Technical Language Precision:**

5: Very High Quality – The language is entirely appropriate for a technical report. All terms are used correctly and precisely, and the style is professional, clear, and coherent, without any errors or ambiguities.

---

### Official Review · Reviewer_WCaa · 2024-12-06
**Recenzja projektu VirtualCockpit**

**Confidence:** 4
**Significance Of Results:** 5
**Overall Quality:** 5

**Compliance With Template:**

5: Very High Quality – The article contains all the required sections, which are written in a very detailed, clear, and error-free manner. The structure is professional and meets expectations, and the content adheres to the highest substantive and formal standards.

**Description Of Results:**

4: High Quality – The results are described in detail and supported by usage examples or evaluations. The description is reliable but may lack full depth of analysis.

**Feedback On Consistency:**

Praca jest napisana prostym, poprawnym i zrozumiałym językiem oraz wolna od kolokwializmów. Znajdują się w niej wszystkie wymagane elementy, które są w zadowalający sposób rozwinięte. Szczególnie wartościowe jest porównianie z innymi pracami i dyskusja różnic w stosunku do zrealizowanego projektu.
Osiągnięte wyniki przedstawiono w sposób poprawny, jedyną uwagę mam względem przyjętych wartości czasu opóźnień. O ile intuicyjnie można je uznać za rozsądne, to warto byłoby odnieść się tutaj do badań i innych artykułów, w których problem zdalnego sterowania pojazdami jest szerzej rozważany i odnieść własne założenia do wartości w nich przedstawianych.
Artykuł omawia wyniki testów w odizolowanym środowisku, a następnie jest mowa o wykorzystaniu elementów sieci 5G. Choć jedno nie wyklucza drugiego warto byłoby sprecyzować w jakim dokładnie środowisku przeprowadzono testy oraz ewentualnie przeprowadzić testy w warunkach zbliżonych do rzeczywistości (np. w realnie działajacej sieci 5G).

**Potential For Development:**

Praca jest bardzo mocno osadzona w kontekście praktycznym. Autorzy wskazują kilka zastosowań własnego projektu, również tych nie związanych bezpośrednio z założonym celem (np. obsługa maszyn budowlanych). W przekonujący sposób przedstawione są potencjalne przypadki użycia, które są przemyślane i odpowiadają problemom realnie występującym w kontekście pojazdów autonomicznym.
Jedynym elementem, który można byłoby rozbudować jest wskazanie koniecznych do wprowadzenia modyfikacji, usprawnień i funkcji, które byłyby wymagane we wdrożeniach w alternatywnych środowiskach.

**Project Nature Evaluation:**

Projekt ma charakter konstrukcyjny i z pewnością spełnia wymogi stawiane projektom inżynierskim. Dobór metod, narzędzi oraz użytych rozwiązań technologicznych jest poprawny i właściwie umotywowany.
Opis techniczny można byłoby rozszerzyć o wykaz elementów sprzętowych użytych do realizacji projektu, np. specyfikację platformy testowej, modele użytych czujników itp., jednocześnie wskazując na ewentualne ograniczenia z tego wynikające.

**Technical Language Precision:**

4: High Quality – The language is appropriate for a technical report. Terminology is used correctly, and statements are precise, with only minor shortcomings that do not affect the overall clarity.

---

### Official Review · Reviewer_tKae · 2024-12-06
**The article is well-written and demonstrates a thorough understanding of the subject matter. It presents a well-thought-out approach to addressing real-world challenges, showcasing innovative solutions with practical applications.**

**Confidence:** 5
**Significance Of Results:** 4
**Overall Quality:** 5

**Compliance With Template:**

5: Very High Quality – The article contains all the required sections, which are written in a very detailed, clear, and error-free manner. The structure is professional and meets expectations, and the content adheres to the highest substantive and formal standards.

**Description Of Results:**

4: High Quality – The results are described in detail and supported by usage examples or evaluations. The description is reliable but may lack full depth of analysis.

**Feedback On Consistency:**

The project description is consistent and well-structured, with a logical flow from problem analysis through the presentation of results to the conclusions. Each section aligns cohesively, ensuring a clear understanding of the challenges addressed and the practical outcomes achieved, but it lacks the inclusion of at least one screenshot or visual representation of the application. This addition would enhance understanding by allowing readers to not only comprehend the features descriptively but also see how the application is practically implemented.

**Potential For Development:**

The article effectively explores promising possibilities for further development and practical applications of its system. By addressing critical scenarios and operational improvements, it demonstrates significant potential to enhance safety, efficiency, and resource management. Its well-rounded approach positions the system as a valuable contribution to advancing autonomous and remote vehicle technologies.

**Project Nature Evaluation:**

The project demonstrates clear characteristics of engineering work by meeting key criteria. It showcases a high level of utility through its practical applications. The use of advanced technical methods, including remote control systems, data streaming, and sensor integration, highlights its reliance on precise and innovative technological solutions. Additionally, the system addresses real-world challenges with a structured and problem-solving approach, emphasizing its engineering foundation.

**Technical Language Precision:**

5: Very High Quality – The language is entirely appropriate for a technical report. All terms are used correctly and precisely, and the style is professional, clear, and coherent, without any errors or ambiguities.

---

### Decision · Program_Chairs · 2024-12-10

Accept (Oral)